# Effects of an oral exercise intervention on pre-frailty or frailty in older people: a randomized clinical trial

## Abstract

**Background** Frailty is often experienced by older adults, which can lead to long-term health problems. We aimed to examine associations with improvements in nutritional status, sarcopenia (age-related loss of skeletal muscle mass and strength), and frailty in four groups with different oral exercise frequencies.

**Methods** We conducted a prospective, parallel multi-arm randomized controlled trial (Japan Registry of Clinical Trials (jRCT) 1062210063) to test the effects of oral exercise on frailty in older adults. Each intervention consisted of a standardized oral exercise protocol including neck exercises, lip exercises, and tongue movements, designed to improve oral function and reduce frailty. The primary outcome was the change in the number of frailty criteria from baseline to follow-up. Individuals aged ≥60 years were screened for frailty status using standardized criteria at the Department of Preventive Dentistry at Okayama University Hospital between October 2022 and December 2023. Those identified as pre-frailty or frailty were eligible and enrolled in the study. After screening 60 individuals, 58 eligible participants were randomly assigned using block randomization to one of four oral exercise frequency groups: 3 times/day & everyday, 3 times/day & 3 days/week, once/day & everyday, and once/day & 3 days/week. A two-way repeated measures analysis of variance was used to evaluate the impact of the four frequencies of oral exercise methods on frailty in older adults. Outcome assessors were blinded; participants were not.

**Results** Here we show the results of the 58 participants. Group sizes are: 3 times/day & everyday ($n = 14$), 3 times/day & 3 days/week ($n = 15$), once/day & everyday ($n = 14$), once/day & 3 days/week ($n = 15$). The trial is completed as planned, and all randomized participants are analyzed. The main effect of time is significant for the number of frailty criteria ($F = 14.803$, $p < 0.001$, partial eta squared = 0.215). The mean changes from baseline to follow-up are $-0.357$ (95% Confidence Interval $-0.787$ to $0.073$) in the 3 times/day & everyday group, $-0.600$ (95% Confidence Interval $-1.255$ to $0.055$) in the 3 times/day & 3 days/week group, $-0.571$ (95% Confidence Interval $-1.379$ to $0.236$) in the once/day & everyday group, and $-0.600$ (95% Confidence Interval $-1.008$ to $-0.192$) in the once/day & 3 days/week group. The main effect of time is also significant for the number of oral hypofunction criteria ($F = 16.456$, $p < 0.001$, partial eta squared = 0.234). No important adverse events or side effects related to the intervention were observed.

**Conclusions** After conducting oral exercises for 3 months on older adults with pre-frailty or frailty, improvements in frailty are observed. Overall, these exercises could be a simple, low-cost way to support healthy aging in the community.

## Plain language summary

Older adults often experience frailty, which can lead to health problems and reduced independence. This study tested whether simple oral exercises, such as moving the tongue and cheeks, can help improve frailty. We randomly assigned 58 older adults with frailty to four groups that performed oral exercises at different frequencies for three months. We found that frailty scores improved in all groups, and the most practical and effective schedule was three times a day, three days a week. These exercises are easy to do at home and may help older adults maintain better health and quality of life. This approach could be a simple, low-cost way to support healthy aging in the community.

✉e-mail: takeuti@md.okayama-u.ac.jp

Frailty refers to an aging-associated state of decline in physical, mental, and social functions[1]. This condition includes a reduction in muscle strength, the deterioration of nutritional status, decreased activity levels, cognitive decline, and increased feelings of loneliness. Frailty is associated with adverse outcomes such as increased mortality and higher rates of hospitalization, disability, and falls[2]. It is a preventable stage that occurs before dependency or the need for advanced medical care, and appropriate interventions can slow its progression.

In cross-sectional and longitudinal studies, oral status has been reported to contribute potentially to the prevention of sarcopenia and frailty through nutritional status. In a cross-sectional study using structural equation modeling, Sawada et al.[1,2] reported that tongue–lip motor function was positively associated with nutritional status, and that nutritional status was negatively associated with frailty. In a review, Moynihan et al.[3] found that tooth loss was associated with an increased risk of sarcopenia, but the data were largely cross-sectional; thus, longitudinal data and intervention studies are needed to clarify whether the association between poor oral function and sarcopenia is causal or bidirectional.

Silva et al.[4] reported that tongue strengthening exercises not only increase muscle strength and thickness and exert a positive effect on swallowing, but also prevent sarcopenia and avoid functional changes[5,6]. Kawamura et al.[7] found that individuals who routinely chewed gum had better oral, cognitive, and physical functions, and thus may contribute to the prevention of frailty.

In another review, Morisaki[8] discussed oral exercise methods for improving oral function based on 14 previous studies and reported that oral exercises should involve a combination of multiple movements and be performed frequently for more than 2 months. However, many questions remain regarding the optimal methods.

Oral function training has been reported to maintain and improve oral function; however, few intervention studies have reported associations between the maintenance and improvement of nutritional status and physical function. Sasajima et al.[9] reported significant improvements in the one-leg standing time with eyes open and the Timed Up and Go tests after 3 months of physical and oral function training. However, they also stated that more detailed studies are needed to confirm the relationship between oral health and physical performance.

To our knowledge, very few intervention studies investigate the relationship between frailty and oral function training alone. In addition, reports discussing the frequency of oral function training feasible for social implementation are lacking. In the present study, we hypothesize that the implementation of oral exercises leads to improvements in nutritional status, sarcopenia, and frailty, and that a minimum frequency of oral exercises is required for these improvements to occur. Given this background, this randomized controlled trial aimed to examine associations with improvements in nutritional status, sarcopenia, and frailty in four groups with different oral exercise frequencies. The results show that oral exercises performed three times per day and three days per week improve frailty status and achieve the highest adherence. These findings suggest that a socially feasible oral exercise routine can contribute to reducing frailty in older adults without disabilities or cognitive impairments.

## Methods
### Study design
This study was planned and conducted as a prospective, parallel multi-arm randomized controlled trial following the CONSORT 2025 checklist. The intervention consisted of an identical oral exercise program across all groups, with only the frequency differing among the four groups. The four intervention groups were designed not only to examine the effect of oral exercises on pre-frailty or frailty improvement, but also to explore feasible implementation strategies for social application. Consequently, the study concentrated on the comparison of practical combinations of oral exercise frequency (per day and per week). Written informed consent was obtained from all participants. This trial is registered in the Japan Registry of Clinical Trials (jRCT; jRCT1062210063) on December 6, 2021 and is completed. The full trial protocol is publicly available in the Japan Registry of Clinical Trials under the registration number jRCT1062210063. There were no important changes in methodology after the trial commencement.

### Participants
The study participants were recruited from participants aged ≥60 years who regularly visited the Department of Preventive Dentistry at Okayama University Hospital. The exclusion criteria included being unable to walk independently or respond to the questionnaire. Participants or members of the public were not involved in the design, conduct, or reporting of this trial.

**Fig. 1 | Flow chart of oral exercise intervention program participants.** This figure shows the participant flow through the study, including screening, randomization into four oral exercise frequency groups (3 times/day & everyday; 3 times/day & 3 days/week; once/day & everyday; once/day & 3 days/week), and inclusion in intention-to-treat (n = 58) and per-protocol analyses (n = 53).

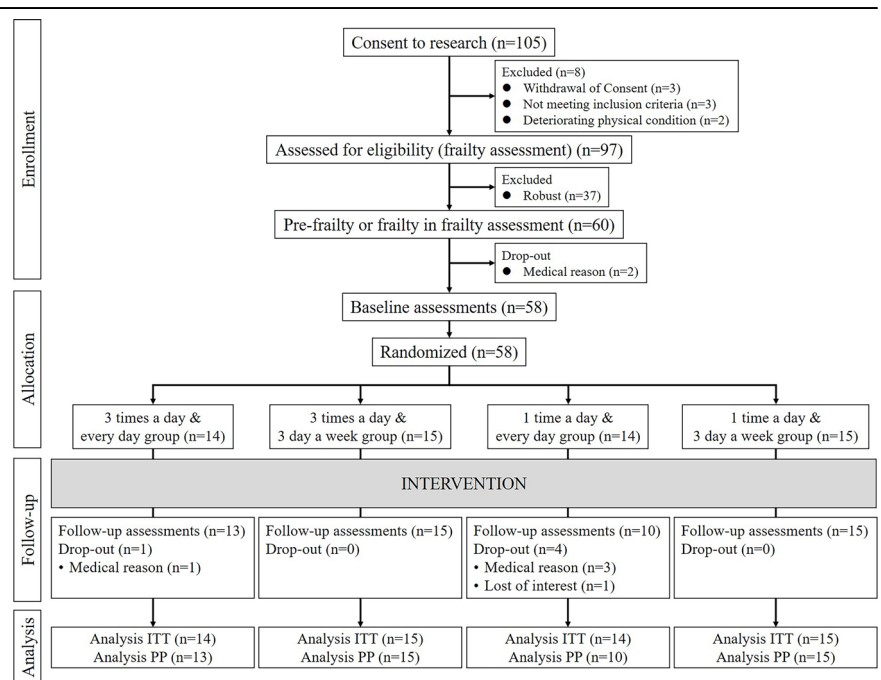

**Fig. 2 | Oral exercises leaflet.** This figure displays the leaflet used in the intervention, illustrating specific oral exercise methods and movements for the oral and surrounding areas. Adapted from free licensed materials provided by "Rihatsubame", in Japanese (https://zaitaku-st.com/), used under an Open Access license.

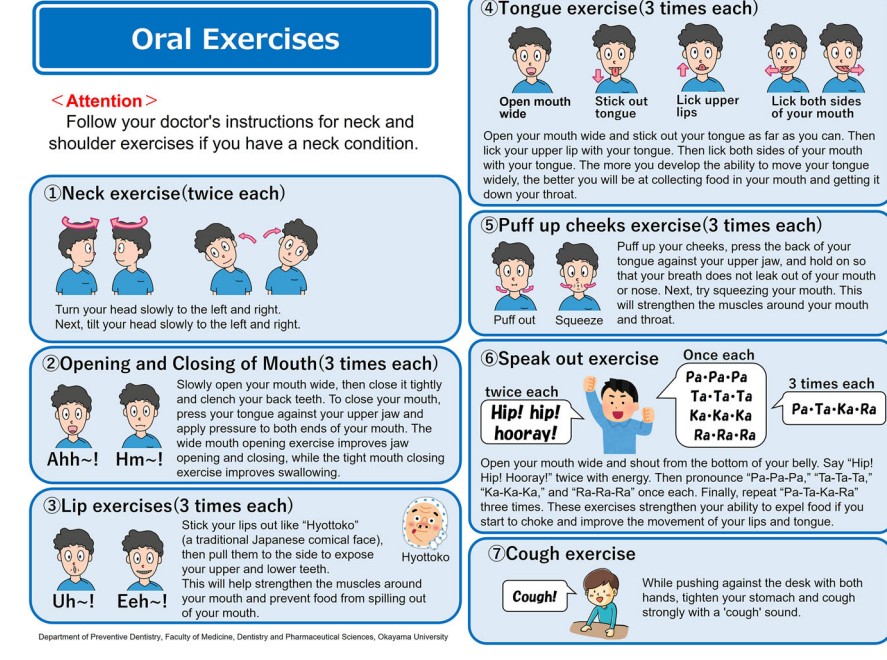

### Recruitment

The recruitment period was from October 11, 2022, to December 22, 2023. After a 3-month intervention period for each participant, reevaluations were conducted from January 13 to March 27, 2024. Follow-up for outcomes of benefits and harms lasted for at least 3 months after the intervention for all participants.

### Participant flow

The details of the participant flow in the present study are shown in Fig. 1. In this study, 105 participants who provided consent were recruited. From the eligibility assessment for frailty, 60 individuals categorized as pre-frailty or frailty were selected and randomly assigned to the following four oral exercise groups: (1) three times a day, seven days a week (3 times/day & everyday; $n = 15$); (2) three times a day, three days a week (3 times/day & 3 days/week; $n = 15$); (3) once a day, seven days a week (once/day & everyday; $n = 15$); and (4) once a day, three days a week (once/day & 3 days/week; $n = 15$).

### Sample size

Sample size was calculated using STATA software (version 19.5; Stata Corp LLC, College Station, TX, USA) based on the primary outcome, namely improvement in frailty after oral exercise intervention. As there were no previous studies on oral exercise alone for frailty, reference values were obtained from studies reporting changes in tongue pressure following oral exercise[10]. Effect sizes were defined as Cohen's d for change scores and converted to Cohen's f for repeated-measures ANOVA. Specifically, the difference in change between the experimental and control groups was $\Delta = 14.6$, with a pooled standard deviation of 4.5 (Cohen's $d \approx 3.24$; Cohen's $f \approx 1.62$). To be conservative, α was set at 0.01, power at 0.95, the within-subject correlation at $r = 0.5$, and the error variance (*varerror*) at $4.5^2$ in Stata's "power repeated" command, yielding a required sample size of 12 participants per group (total N = 48). Allowing for an anticipated dropout rate of 20%, a total of 60 participants were recruited. The same sample size calculation was not applied to secondary outcomes, which were not used for power calculations.

The criteria for discontinuing the research were as follows: (1) if the research participant requested to withdraw from the study or revoked consent; (2) if the entire research project was discontinued; (3) if the participant engaged in or had participated in oral exercise programs organized by local authorities during the study period; (4) if the participant started or had started new exercises or physical activities during the study period; and

(5) for any other reason deemed appropriate by the research supervisor to terminate the study.

### Randomization and implementation

Participants were randomly assigned to one of the four groups using simple randomization based on an Excel-generated random number table, with an allocation ratio of 1:1:1:1 and no restrictions (i.e., no stratification or blocking). The random sequence was created by N.T., and group assignment was performed by N.S. Allocation concealment was maintained by storing the randomization list in a secured binder to be consulted only at the time of assignment. The personnel who enrolled participants and those who assigned participants to interventions both had potential access to the random allocation sequence; however, access was restricted by protocol and controlled operationally.

### Interventions

Participants received standardized oral exercise instructions after a baseline examination. A trained dental hygienist (I.S.) provided face-to-face guidance using individual leaflets (Fig. 2). Participants were instructed to record on a calendar whether they performed the oral exercises as instructed for the prescribed frequency; the calendar was collected during the reevaluation (Fig. 3). All participants were directed to engage in the oral exercises for a duration of 3 months. Participants were randomly assigned to one of four exercise frequencies: 3 times/day & everyday; once/day & everyday; 3 times/day & 3 days/week; and once/day & 3 days/week. A random number table was used for the assignments, ensuring a ratio of 1:1:1:1 for random allocation to one of the four groups.

Implementation was monitored using these calendars, and the implementation rate was calculated as the percentage of completed exercises relative to the prescribed number. Fidelity was maintained by providing identical instructions and materials across all groups, with the only difference being exercise frequency. No additional interventions beyond the assigned oral exercise program were provided.

### Outcomes

The primary outcome was the change in the number of frailty criteria from baseline to follow-up, and the primary analysis was conducted in the intention-to-treat population. The demographic characteristics of the enrolled participants were age and sex. The secondary outcomes included

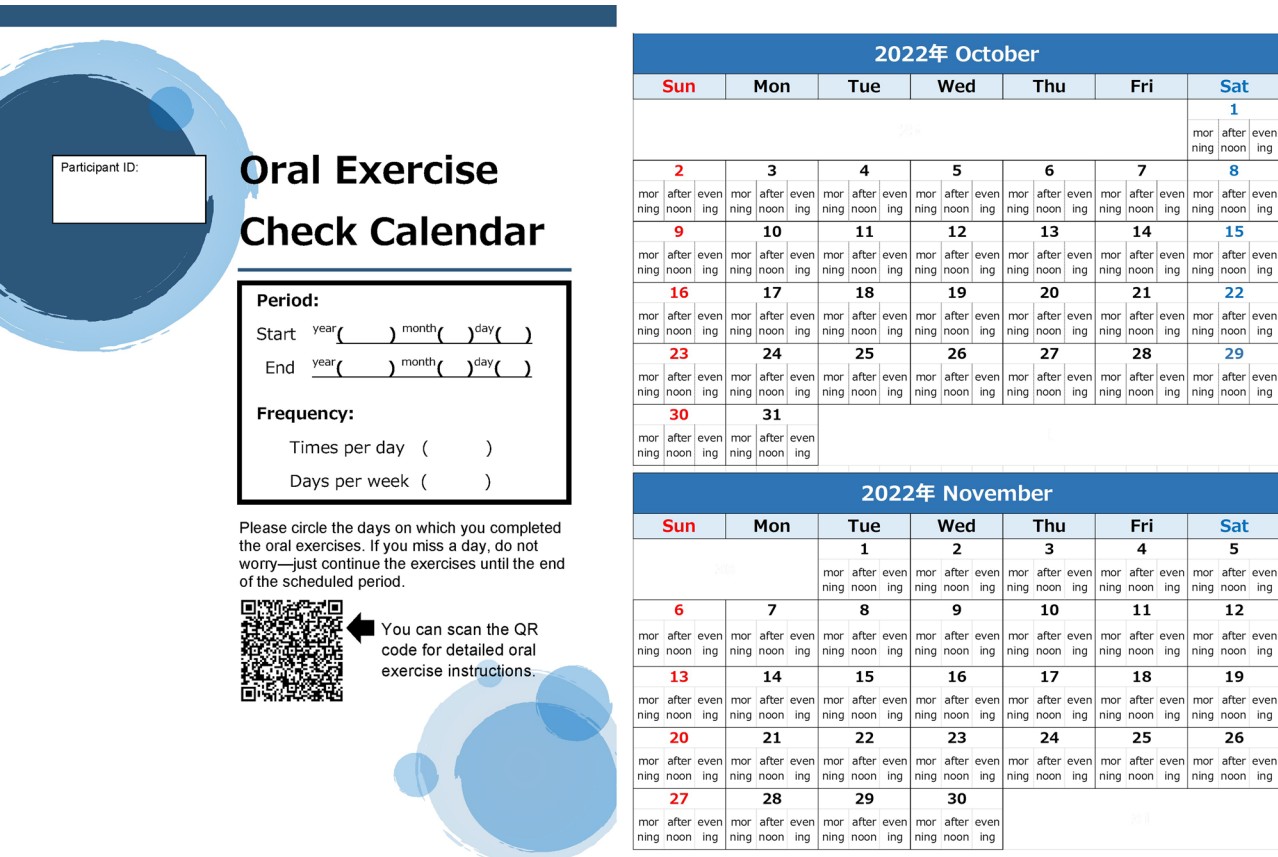

**Fig. 3 | Calendar for checking the implementation of oral exercises.** This figure shows the calendar used to record the implementation rate of the oral exercise program. The sheet includes a QR code linking to detailed exercise instructions.

body mass index (BMI), skeletal muscle mass, appendicular skeletal muscle mass, SMI, nutritional status, mental health status, social frailty, and oral status, including oral function.

### Assessment of frailty
The frailty assessment utilized the revised Japanese version of the Cardiovascular Health Study criteria (revised J-CHS criteria)[11] for defining frailty. Globally, the most commonly used diagnostic criteria are those proposed by Fried, known as the CHS criteria. For screening frailty, these criteria summarize the signs expressed because of aging-associated functional decline into a phenotypic model.

In the revised J-CHS criteria, frailty is defined by five diagnostic criteria: (1) weight loss, (2) fatigue, (3) decreased physical activity, (4) slowness (delayed walking speed), and (5) weakness (muscle strength reduction). If three or more criteria are met, the individual is classified as frailty; if one to two criteria are met, the individual is classified as pre-frailty; and if no criteria are met, the individual is considered robust. For the measurement of weakness (muscle strength reduction), grip strength was assessed using a handgrip dynamometer (TOEI LIGHT CO., LTD., Saitama, Japan).

### Physical status
**BMI.** BMI was calculated using a body composition meter with a stature meter (BH-300A-N, TANITA, Tokyo, Japan) to measure height (m) and weight (kg), and is defined as the value obtained by dividing weight by the square of height.

### Skeletal muscle mass, appendicular skeletal muscle mass, and SMI
Skeletal muscle mass, appendicular skeletal muscle mass, and SMI measurements were conducted using a body composition analyzer that

employs two frequencies and an eight-electrode system with a grip (InnerScan DUAL RD-804L; TANITA, Tokyo, Japan). Skeletal muscle mass was measured for the whole body and five specific regions (left arm, right arm, left leg, right leg, and trunk). Appendicular skeletal muscle mass was defined as the sum of the muscle mass in the left arm, right arm, left leg, and right leg. SMI is an indicator of whether the muscle mass in the arms and legs is excessive relative to height, calculated by dividing appendicular skeletal muscle mass (kg) by the square of height (m).

### Nutritional status
The Mini Nutritional Assessment (MNA)[12] was used to evaluate nutritional status. The MNA is a tool designed to assess the nutritional status of older individuals aged ≥65 years. It is composed of 18 items, including six screening items (food intake, weight loss, mobility, acute illness within the last 3 months, psychological problems, and BMI) and 12 assessment items (independence in daily living, taking four or more medications, presence of pain or ulcers, number of meals per day, protein intake, daily consumption of at least two servings of fruits and vegetables, hydration, assistance with meals, self-assessment of nutrition, self-assessment of health status, upper arm circumference, and calf circumference), with a higher score indicating better nutritional status.

### Mental health status
Mental health status was assessed using the Geriatric Depression Scale-15 Japanese version[13] (GDS-15-J). The GDS-15-J is a validated scale for evaluating depression in older adults composed of 15 questions that can be answered with a simple "yes" or "no". A total score of ≥ 6 indicates the presence of depressive tendencies[13,14]. The data for this study were based on the total scores.

## Social frailty

Social frailty was evaluated based on the criteria established by Makizako et al.[15]. The evaluation includes five items: "Living alone (yes: 1 point)", "Frequency of going out has decreased compared with last year (yes: 1 point)", "Visiting friends' homes (no: 1 point)", "Feeling that I am helpful to family or friends (no: 1 point)", and "Having daily conversations with someone (no: 1 point)". A total score of $\geq 2$ points is classified as social frailty. The data for this study were based on the total scores.

## Oral status

Oral status was examined based on the present number of teeth, number of functional teeth, bacteria count on the dorsal surface of the tongue, oral moisture status, occlusal force, tongue–lip motor function, tongue pressure, masticatory function, and swallowing function. The oral status examination included both oral cavity and oral function assessments. The oral examination was assessed in regard to the number of present and functional teeth. Oral function was examined in terms of bacterial count on the dorsal surface of the tongue, oral moisture status, maximum occlusal force, tongue–lip motor function, maximum tongue pressure, masticatory function, and swallowing function. The oral examination was conducted by N.T., D.E., and M.M., while the oral function examination was performed by S.I and N.S.

## Number of present and functional teeth

The number of present teeth included natural teeth, restored teeth, and untreated teeth, and did not include artificial teeth or retained roots. The number of teeth present was defined as the number of teeth erupting into the oral cavity, even partially. The number of functional teeth included natural and prosthetic teeth, dentures, and implants[16]. The remaining roots were excluded.

## Bacterial count on the dorsal surface of the tongue

The bacterial count on the dorsal surface of the tongue was measured using a simple oral bacterial quantification device (Bacterial Counter; Panasonic Healthcare, Tokyo, Japan). A sterilized swab was attached to a constant-pressure specimen collection device, and the central part of the dorsal tongue was swabbed back and forth for ~1 cm in length three times to collect the sample. Subsequently, the swab, a dedicated sensor chip, and a disposable cup containing 5.0 mL of distilled water were inserted into the designated position of the bacterial counter device. Bacteria in the liquid were collected onto the electrodes through dielectrophoresis, and the change in impedance was measured to quantify the bacterial concentration (CFU/mL) in the sample, allowing for a quantitative assessment of the bacterial count[17].

## Oral moisture status

Oral moisture status was measured using an oral moisture meter (Mucus; Life, Koshigaya, Japan) placed at the center of the tongue[18], and the average of two measurements was taken as the final value. Participants were instructed not to eat, drink, or rinse their mouths for at least 30 min prior to the measurement. A lower value indicates a lower oral moisture level.

## Maximum occlusal force

The DENTAL PRESCALE II (GC Corporation, Tokyo, Japan), which consists of two layers of polyethylene terephthalate film and a color-developing material sandwiched with numerous microcapsules, was used to measure occlusal force. This device was utilized alongside a pressure-sensitive film and an occlusal force analysis device (GC Corporation). Subsequently, the films were scanned using occlusal force analysis software to measure the occlusal force. The unit of measurement for the occlusal force was N, and the pressure filter function of the occlusal force measuring device was employed. Measurements were taken with the dentures in place, and the average of two measurements was used for analysis.

## Tongue-lip motor function

The evaluation of tongue-lip motor function was conducted using an automatic measurement device (Kenko-kun Handy; Takei Scientific Instruments Co., Ltd., Tokyo, Japan). The motor abilities of the lips, anterior tongue, and posterior tongue were assessed by recording the number of repetitions per second for the monosyllables /pa/, /ta/, and /ka/. Each syllable was pronounced for a total of 5 seconds, and the number of syllables produced per second was evaluated. Each measurement was conducted twice, and the average values were used for analysis.

## Maximum tongue pressure

The evaluation of tongue pressure was conducted using the JMS Tongue Pressure Measurement Device (TPM-01; JMS, Hiroshima, Japan). This device objectively assesses tongue strength using maximum tongue pressure as an indicator. A balloon-type probe was used for measurement; after automatically pressurizing the balloon to a predetermined value outside the oral cavity, the balloon of the tongue pressure probe was placed on the tongue inside the oral cavity. The hard ring of the tongue pressure probe was fixed by biting down with the front teeth, and the balloon inside the oral cavity was compressed against the hard palate with the maximum force of the tongue to measure the maximum tongue pressure. Regarding data collection, practice was conducted until the above procedure could be performed smoothly, followed by two measurements, and the average value was taken as the tongue pressure value.

## Masticatory function

Masticatory function was measured using the masticatory ability testing system (GLUCO SENSOR GS-2, GC Corporation). A cylindrical gummy jelly (GLUCOLUMN; GC Corporation) with a diameter of 14 mm, a height of 8 mm, and a weight of ~2 g was chewed on the habitual side for 20 s, and the glucose elution amount was calculated. To calculate the glucose elution amount, after chewing the gummy jelly, the participant held 10 mL of water in the mouth and spat it into a filter cup, where the glucose concentration in the filtrate was measured using a glucose meter. Measurements were performed twice, and the average value was used for analysis.

## Swallowing function

Assessment of swallowing function was conducted using the Japanese version of the Eating Assessment Tool-10 (EAT10)[19] and the repetitive saliva swallowing test (RSST). The EAT10 consists of a questionnaire with 10 items designed to screen for swallowing disorders. Each question is rated on a scale from 0 (no problem) to 4 (severe problem), resulting in a total score ranging from 0 to 40 points. A score of $\geq 3$ is considered to indicate a "risk of aspiration". In this study, the total score was used for analysis.

The RSST is a swallowing disorder screening method that measures how many times a person can perform dry swallowing in 30 s while palpating the thyroid cartilage. Being unable to perform three dry swallows is considered to indicate an increased risk for swallowing disorders[20]. This method does not require special equipment and poses no risk of aspiration during the test, and thus, it is widely used in many care facilities in Japan. In this study, the total score was used for analysis.

## Oral hypofunction

The Japanese Society of Gerodontology proposed "oral hypofunction" as the stage at which recovery can be expected by performing dental treatment before oral dysfunction occurs. Therefore, we developed a hypothesis that oral frailty and hypofunction emerge during the process toward oral dysfunction among the various declines in ability. In addition, as a starting point for discussing this problem, we presented criteria for diagnosing oral hypofunction.

Seven conditions were set to diagnose poor oral function: poor oral hygiene, xerostomia, poor bite strength, poor lingual–labial motor function, poor tongue pressure, poor masticatory function, and poor swallowing function. Furthermore, based on the results of a study[21] conducted at Fujita Health University Hospital, "oral hypofunction" was defined as a condition

that meets three or more of these diagnostic criteria. The number of criteria met for these seven items was designated as the "number of criteria for oral hypofunction".

### Systemic disease

A questionnaire was used to record the participants' medical history and number of medications taken. The questionnaire requested information on the following conditions that have been linked to frailty: stroke[22], heart disease[23], respiratory disease[24], hypertention[25], diabetes mellitus[26], kidney disease[27], knee osteoarthritis[28], osteoporosis[29], rheumatoid arthritis[30], Parkinson's disease[30], anemia[31], and hearing dysfunction[24].

### Implementation rate

The implementation rate was calculated by dividing the total number of actual oral exercises performed by the total number of exercises if all had been conducted, then multiplying by 100. The total number of exercises for each group was as follows: 252 times for the 3 times/day & everyday group; 108 times for the 3 times/day & 3 days/week group; 84 times for the once/day & everyday group; and 12 times for the once/day & 3 days/week group.

### Blinding

This study utilized a single-blind design. Each assessor evaluated regardless of group. The evaluators were not informed of which group the subjects were assigned to, only one research team member maintained information about study group assignment and communication with subjects occurred only through that one person. The interventions were similar in content and instructions; the only difference was the frequency of oral exercise.

### Statistics and reproducibility

Descriptive statistics were used to summarize the participants' demographics, baseline characteristics, and study outcomes. The normality of data distribution was assessed using the Kolmogorov-Smirnov test. As most variables, except for age, skeletal muscle mass, and tongue pressure, were not normally distributed, nonparametric tests were used for analysis.

All data are presented as means and standard deviations. Comparisons among the four groups at baseline were conducted using the Kruskal-Wallis test. Comparisons between baseline and follow-up for each group were performed using the Wilcoxon signed-rank test. The primary analysis was based on the intention-to-treat (ITT) population; per-protocol (PP) analysis was performed as a sensitivity analysis. Confidence intervals for the mean difference between baseline and reassessment were calculated with 95% confidence intervals for the primary outcome and 99.8% confidence intervals for the other 22 items to control for multiplicity. MCID for each measure was calculated using half the standard deviation of the difference between baseline and reassessment (reassessment value minus baseline value)[32].

We conducted a two-way repeated measures ANOVA to evaluate the effects of each group (3 times/day & everyday, 3 times/day & 3 days/week, once/day & everyday, and once/day & 3 days/week) and time (baseline and follow-up) on number of frailty criteria, SMI, number of present teeth, number of functional teeth, number of oral hypofunction criteria, MNA, Social frailty score, GDS15. Mean differences for the main outcome number of frailty criteria were calculated with 95% confidence intervals. The 99.3% confidence interval was calculated for the following seven secondary outcomes (SMI, number of present teeth, number of functional teeth, number of oral hypofunction criteria, MNA, social frailty score, and GDS15) to account for multiplicity using Bonferroni correction. Partial eta squared ($\eta p^2$) was utilized to assess the effect size (ES), with values of $0.01 < \eta p^2 < 0.06$ indicating a small effect, $0.06 \leq \eta p^2 \leq 0.14$ representing a medium effect, and $\eta p^2 > 0.14$ denoting a large effect. Planned pairwise comparisons were performed using the Bonferroni post hoc correction to identify differences. Cohen's d was employed to calculate the ES for the post hoc comparisons, which was categorized as trivial (<0.20), small (0.20–0.49), moderate (0.50–0.79), or large ($\geq 0.80$). An additional analysis using generalized linear models (logistic regression) was conducted for categorical outcomes

(frailty[33], EAT-10[34], social frailty[15], and GDS-15[35]) to confirm the robustness of the findings. This analysis was performed post hoc.

The implementation rate was calculated by dividing the actual number of oral exercises performed by the number of times they should have been performed, and then multiplying by 100. The differences in implementation rates were analyzed using the Kruskal-Wallis test with Bonferroni correction.

All other statistical analyses were performed using SPSS (version 26.0; IBM, Armonk, NY, USA). The overall significance level was set at $p < 0.05$ for the primary outcome, and Bonferroni correction was applied for multiple comparisons to control the family-wise error rate. Although this adjustment was noted in the protocol or SAP, it was implemented post hoc to ensure rigorous control of type I error across multiple comparisons. For oral moisture status, maximum occlusal force, tongue–lip motor function, tongue pressure, and masticatory function, each measurement was performed twice at both baseline and follow-up, and the average of the two values was used for analysis. All other outcome measures were assessed once per participant at each time point. Each participant was considered as a single experimental unit, and no technical replicates beyond the repeated measurements were used. Missing data were handled as follows: For participants with two scheduled measurements, if one measurement was missing, the available measurement was used in the analysis rather than calculating an average. No imputation was performed.

### Ethical considerations

This study was conducted in accordance with the tenets of the Declaration of Helsinki, and procedures involving human participants were approved by the ethics committee of Okayama University Graduate School of Medicine, Dentistry, and Pharmaceutical Sciences and Okayama University Hospital (approval No.: 1803-038).

## Results

### Number analyzed

A total of 58 participants who began the trial were included in the ITT analysis, and 53 who adhered to their assigned protocol were involved in the PP analysis.

### Baseline characteristics

The characteristics of the participants at baseline, including demographic, clinical, physical, oral function, and nutritional variables, are shown in Table 1. The full version of this table, including all variables, is provided in Supplementary Table 1. No significant differences were observed among the four groups for all items.

### Outcomes and estimation

There were no changes to trial outcomes after the trial commenced. Table 2 shows the baseline and follow-up outcomes, including the means ± standard deviations, within-group comparisons, and between-group comparisons. The confidence intervals (CI) for the mean differences (95% CI for the frailty criterion and 99.8% CI for the other 22 items to account for multiplicity) and the minimum clinically important difference (MCID) in each group are provided in Supplementary Table 2. There were no significant differences between the four groups at baseline.

### Primary outcome

A two-way repeated measures analysis of variance (ANOVA) was used to evaluate the impact of the four frequencies of oral exercise methods on frailty in older adults (Table 3). The results regarding the analysis of effect sizes and CI for the mean differences (95% CI for the frailty criterion and 99.3% CI for the other 7 items to account for multiplicity) are shown in Table 3. For the primary outcome (number of frailty criteria), the estimated mean differences from baseline to follow-up in the ITT population were −0.357 (95% CI, −0.787 to 0.073) in the group performing oral exercises three times per day every day (3 times/day & everyday group), −0.600 (95% CI, −1.255 to 0.055) in the 3 times/day & 3 days/week group, −0.571 (95%

**Table 1 | Participants' characteristics (intention-to-treat analysis, Simplified Version)**

| Variables | Total | 3 times/day & everyday (*n* = 14) | 3 times/day & 3 days/week (*n* = 15) | once/day & everyday (*n* = 14) | once/day & 3 days/week (*n* = 15) | *p* value[c] |
|---|---|---|---|---|---|---|
| Age (y) | 79.71 ± 6.39[a] | 76.86 ± 4.75 | 78.87 ± 5.00 | 81.64 ± 6.05 | 81.40 ± 8.42 | 0.147 |
| Sex (Male) | 12 (20.69)[b] | 3 (21.43) | 4 (26.67) | 2 (14.28) | 3 (20.00) | 0.877 |
| **Number of frailty criteria** | 2.17 ± 0.99 | 2.07 ± 1.00 | 2.13 ± 0.83 | 2.36 ± 1.28 | 2.13 ± 0.92 | 0.949 |
| Grip strength (kg) | 18.73 ± 6.39 | 19.12 ± 4.10 | 20.82 ± 7.46 | 15.79 ± 7.23 | 19.01 ± 5.72 | 0.207 |
| Walking speed (m/s) | 1.04 ± 0.27 | 1.03 ± 0.25 | 1.10 ± 0.25 | 1.04 ± 0.32 | 0.98 ± 0.27 | 0.659 |
| **SMI (kg/m$^2$)** | 6.13 ± 0.81 | 5.99 ± 0.60 | 6.34 ± 0.80 | 6.05 ± 0.94 | 6.12 ± 0.91 | 0.658 |
| BMI (kg/m$^2$) | 22.24 ± 4.13 | 21.28 ± 5.13 | 23.44 ± 3.01 | 21.42 ± 3.22 | 23.31 ± 4.56 | 0.191 |
| Skeletal muscle mass | 33.45 ± 6.59 | 32.64 ± 4.94 | 34.81 ± 6.73 | 32.73 ± 7.18 | 33.81 ± 7.09 | 0.750 |
| Appendicular skeletal muscle mass | 14.44 ± 3.17 | 14.09 ± 2.48 | 15.19 ± 3.28 | 14.21 ± 3.89 | 14.21 ± 3.10 | 0.757 |
| Number of present teeth (n) | 19.88 ± 7.47 | 20.50 ± 7.69 | 19.80 ± 7.70 | 19.79 ± 6.23 | 19.47 ± 8.72 | 0.955 |
| Number of functional teeth (n) | 26.52 ± 3.24 | 25.93 ± 2.23 | 27.07 ± 2.15 | 25.36 ± 5.27 | 27.60 ± 2.10 | 0.307 |
| **Number of Oral hypofunction criteria** | 3.50 ± 1.41 | 3.00 ± 1.11 | 3.40 ± 1.40 | 4.14 ± 1.83 | 3.47 ± 1.06 | 0.200 |
| Bacterial count on the dorsal surface of the tongue (10$^6$) (cfu/ml) | 6.11 ± 6.52 | 5.86 ± 5.65 | 5.81 ± 8.06 | 6.57 ± 7.14 | 6.21 ± 5.54 | 0.757 |
| Oral moisture status | 25.73 ± 3.55 | 25.93 ± 2.95 | 25.06 ± 5.45 | 25.77 ± 2.62 | 26.16 ± 2.52 | 0.954 |
| Maximum occlusal force (N) | 468.49 ± 319.14 | 487.25 ± 345.84 | 437.76 ± 282.77 | 394.88 ± 309.85 | 550.40 ± 347.16 | 0.519 |
| Tongue and lip motor function | | | | | | |
| /pa/sound (times/s) | 5.71 ± 0.96 | 5.58 ± 1.16 | 6.01 ± 0.79 | 5.47 ± 1.08 | 5.75 ± 0.81 | 0.478 |
| /ta/sound (times/s) | 5.61 ± 0.98 | 5.51 ± 1.13 | 5.91 ± 0.65 | 5.40 ± 1.13 | 5.61 ± 0.99 | 0.779 |
| /ka/sound (times/s) | 5.24 ± 0.96 | 5.35 ± 1.04 | 5.44 ± 0.68 | 5.03 ± 1.02 | 5.15 ± 1.11 | 0.736 |
| Maximum tongue pressure (kPa) | 22.44 ± 9.00 | 25.34 ± 7.50 | 24.80 ± 10.32 | 17.96 ± 7.25 | 21.56 ± 9.34 | 0.095 |
| Masticatory function (mg/dL) | 211.58 ± 85.28 | 209.39 ± 95.28 | 227.30 ± 106.98 | 185.09 ± 69.31 | 222.63 ± 64.58 | 0.554 |
| EAT10 | 4.07 ± 5.67 | 2.36 ± 3.95 | 4.67 ± 7.35 | 6.00 ± 6.46 | 3.21 ± 3.81 | 0.521 |
| RSST (times) | 3.09 ± 1.84 | 3.57 ± 2.38 | 2.08 ± 1.82 | 3.14 ± 1.29 | 2.87 ± 1.81 | 0.618 |
| **MNA** | 24.11 ± 3.89 | 23.86 ± 3.24 | 24.83 ± 3.96 | 23.68 ± 4.90 | 24.03 ± 3.63 | 0.755 |
| **Social frailty score** | 1.98 ± 1.22 | 1.50 ± 0.65 | 1.60 ± 1.12 | 2.29 ± 1.27 | 2.53 ± 1.46 | 0.142 |
| **GDS15** | 4.28 ± 3.52 | 3.00 ± 2.32 | 3.80 ± 2.93 | 4.36 ± 3.59 | 5.87 ± 4.50 | 0.322 |
| Stroke | 1 (1.72) | 0 (0.00) | 0 (0.00) | 0 (0.00) | 1 (6.67) | 0.405 |
| Heart disease | 5 (8.62) | 1 (7.14) | 2 (13.33) | 1 (7.14) | 1 (6.67) | 0.903 |
| Respiratory disease | 5 (8.62) | 3 (21.43) | 1 (6.67) | 1 (7.14) | 0 (0.00) | 0.218 |
| Hypertension | 25 (43.10) | 5 (35.71) | 5 (33.33) | 7 (50.00) | 8 (53.33) | 0.613 |
| Diabetes mellitus | 7 (12.07) | 5 (35.71) | 0 (0.00) | 0 (0.00) | 2 (13.33) | 0.010 |
| Kidney disease | 2 (3.45) | 2 (14.29) | 0 (0.00) | 0 (0.00) | 0 (0.00) | 0.089 |
| Knee osteoarthritis | 4 (6.90) | 1 (7.14) | 1 (6.67) | 1 (7.14) | 1 (6.67) | 1.000 |
| Osteoporosis | 10 (17.24) | 4 (28.57) | 1 (6.67) | 3 (21.43) | 2 (13.33) | 0.429 |
| Rheumatoid arthritis | 4 (6.90) | 1 (7.14) | 0 (0.00) | 2 (14.28) | 1 (6.67) | 0.512 |
| Parkinson's disease | 2 (3.45) | 0 (0.00) | 0 (0.00) | 1 (7.14) | 1 (6.67) | 0.557 |
| Anemia | 4 (6.90) | 1 (7.14) | 2 (13.33) | 0 (0.00) | 1 (6.67) | 0.571 |
| Hearing dysfunction | 1 (1.72) | 0 (0.00) | 1 (6.67) | 0 (0.00) | 0 (0.00) | 0.405 |

This table summarizes baseline demographic, clinical, physical, oral function, and nutritional characteristics of participants in the four oral exercise frequency groups under the intention-to-treat analysis. Continuous variables are presented as mean ± standard deviation, and categorical variables as number (percentage). *p* values are based on comparisons among groups. Bold variables indicate the main outcome measures: Number of frailty criteria, SMI, Number of present teeth, Number of functional teeth, Number of oral hypofunction criteria, MNA, Social frailty score, and GDS15.
*BMI* body mass index, *SMI* skeletal mass index, *EAT10* The 10-item Eating Assessment Tool, *RSST* repetitive saliva swallowing test, *MNA* Mini Nutritional Assessment, *GDS* Geriatric depression scale.
[a]Average ± standard deviation.
[b]*n* (%).
[c]*p* values for group comparisons using Kruskal–Wallis test for continuous variables and chi-square test for categorical variables.

## Table 2 | The baseline and follow-up outcomes in the each groups (intention-to-treat analysis, Simplified Version)

| Variables | 3 times/day & everyday (n = 14) | | | 3 times/day & 3 days/week (n = 15) | | | once/day & everyday (n = 14) | | | once/day & 3 days/week (n = 15) | | | p value[b] |
|---|---|---|---|---|---|---|---|---|---|---|---|---|---|
| | baseline | Follow-Up | p value[a] | baseline | Follow-Up | p value | baseline | Follow-Up | p value | baseline | Follow-Up | p value | |
| **Number of frailty criteria** | 2.07 ± 1.00[c] | 1.71 ± 0.91 | 0.096 | 2.13 ± 0.83 | 1.53 ± 0.99 | 0.070 | 2.36 ± 1.28 | 1.79 ± 1.25 | 0.155 | 2.13 ± 0.92 | 1.53 ± 1.25 | 0.013 | 0.949 |
| Grip strength (kg) | 19.12 ± 4.10 | 19.41 ± 3.24 | 0.345 | 20.82 ± 7.46 | 21.09 ± 7.66 | 0.755 | 15.79 ± 7.23 | 16.35 ± 7.68 | 0.414 | 19.01 ± 5.72 | 18.55 ± 5.04 | 0.211 | 0.207 |
| Walking speed (m/s) | 1.03 ± 0.25 | 1.03 ± 0.35 | 0.917 | 1.10 ± 0.25 | 1.15 ± 0.47 | 0.460 | 1.04 ± 0.32 | 1.02 ± 0.27 | 0.799 | 0.98 ± 0.27 | 1.02 ± 0.38 | 0.280 | 0.659 |
| **SMI (kg/m²)** | 5.99 ± 0.60 | 5.98 ± 0.47 | 0.308 | 6.34 ± 0.80 | 6.42 ± 0.72 | 0.152 | 6.05 ± 0.94 | 6.02 ± 0.94 | 0.508 | 6.12 ± 0.91 | 6.20 ± 0.75 | 0.730 | 0.658 |
| BMI (kg/m²) | 21.28 ± 5.13 | 21.07 ± 5.09 | 0.055 | 23.44 ± 3.01 | 23.49 ± 2.99 | 0.691 | 21.42 ± 3.22 | 21.19 ± 3.04 | 0.047 | 23.31 ± 4.56 | 23.23 ± 4.44 | 0.650 | 0.191 |
| Skeletal muscle mass | 32.64 ± 4.94 | 32.68 ± 4.65 | 0.824 | 34.81 ± 6.73 | 34.83 ± 6.10 | 0.382 | 32.73 ± 7.18 | 32.64 ± 7.17 | 0.646 | 33.81 ± 7.09 | 33.80 ± 6.51 | 0.842 | 0.750 |
| Appendicular skeletal muscle mass | 14.09 ± 2.48 | 14.04 ± 2.13 | 0.455 | 15.19 ± 3.28 | 15.37 ± 3.10 | 0.124 | 14.21 ± 3.89 | 14.16 ± 3.89 | 0.678 | 14.21 ± 3.10 | 14.31 ± 2.58 | 0.730 | 0.757 |
| **Number of present teeth (n)** | 20.50 ± 7.69 | 20.43 ± 7.76 | 0.317 | 19.80 ± 7.70 | 19.80 ± 7.70 | 1.000 | 19.79 ± 6.23 | 19.79 ± 6.23 | 1.000 | 19.47 ± 8.72 | 19.47 ± 8.72 | 1.000 | 0.955 |
| **Number of functional teeth (n)** | 25.93 ± 2.23 | 25.86 ± 2.18 | 0.317 | 27.07 ± 2.15 | 27.07 ± 2.15 | 1.000 | 25.36 ± 5.27 | 25.36 ± 5.27 | 1.000 | 27.6 ± 2.10 | 27.6 ± 2.10 | 1.000 | 0.307 |
| **Number of Oral hypofunction criteria** | 3.00 ± 1.11 | 2.50 ± 1.16 | 0.070 | 3.40 ± 1.40 | 3.07 ± 1.49 | 0.374 | 4.14 ± 1.83 | 3.50 ± 1.87 | 0.084 | 3.47 ± 1.06 | 2.53 ± 1.13 | 0.003 | 0.200 |
| Bacterial count on the dorsal surface of the tongue (10⁶)(cfu/ml) | 5.86 ± 5.65 | 5.72 ± 4.35 | 0.969 | 5.81 ± 8.06 | 6.49 ± 7.97 | 0.600 | 6.57 ± 7.14 | 9.44 ± 8.39 | 0.093 | 6.21 ± 5.54 | 11.63 ± 16.35 | 0.307 | 0.757 |
| Oral moisture status | 25.93 ± 2.95 | 29.07 ± 1.01 | 0.005 | 25.06 ± 5.45 | 27.97 ± 2.15 | 0.088 | 25.77 ± 2.62 | 27.69 ± 2.68 | 0.047 | 26.16 ± 2.52 | 29.19 ± 2.01 | 0.003 | 0.954 |
| Maximum occlusal force (N) | 487.25 ± 345.84 | 413.40 ± 378.06 | 0.249 | 437.76 ± 282.77 | 435.05 ± 270.54 | 0.733 | 394.88 ± 309.85 | 418.45 ± 278.61 | 0.445 | 550.40 ± 347.16 | 582.56 ± 298.27 | 0.532 | 0.519 |
| Tongue and lip motor function | | | | | | | | | | | | | |
| /pa/sound (times/s) | 5.58 ± 1.16 | 5.60 ± 1.13 | 0.674 | 6.01 ± 0.79 | 5.99 ± 1.10 | 0.944 | 5.47 ± 1.08 | 5.42 ± 1.29 | 0.812 | 5.75 ± 0.81 | 5.57 ± 0.99 | 0.379 | 0.478 |
| /ta/sound (times/s) | 5.51 ± 1.13 | 5.45 ± 1.23 | 1.000 | 5.91 ± 0.65 | 5.94 ± 0.65 | 0.573 | 5.40 ± 1.13 | 5.48 ± 1.20 | 0.440 | 5.61 ± 0.99 | 5.61 ± 0.93 | 0.283 | 0.779 |
| /ka/sound (times/s) | 5.35 ± 1.04 | 5.19 ± 1.14 | 0.431 | 5.44 ± 0.68 | 5.56 ± 0.64 | 0.277 | 5.03 ± 1.02 | 4.97 ± 1.18 | 0.953 | 5.15 ± 1.11 | 5.18 ± 1.05 | 0.824 | 0.736 |
| Maximum tongue pressure (kPa) | 25.34 ± 7.50 | 24.89 ± 7.85 | 0.650 | 24.80 ± 10.32 | 24.37 ± 9.87 | 0.691 | 17.96 ± 7.25 | 17.76 ± 6.72 | 0.721 | 21.56 ± 9.34 | 25.57 ± 9.73 | 0.394 | 0.095 |
| Masticatory function (mg/dL) | 209.39 ± 95.28 | 243.82 ± 120.76 | 0.173 | 227.30 ± 106.98 | 227.60 ± 69.39 | 0.865 | 185.09 ± 69.31 | 257.39 ± 134.86 | 0.074 | 222.63 ± 64.58 | 255.86 ± 106.24 | 0.496 | 0.554 |
| EAT10 | 2.36 ± 3.95 | 2.71 ± 4.41 | 0.288 | 4.67 ± 7.35 | 3.87 ± 6.31 | 0.438 | 6.00 ± 6.46 | 4.36 ± 6.40 | 0.028 | 3.21 ± 3.81 | 1.87 ± 2.56 | 0.139 | 0.521 |
| RSST (times) | 3.57 ± 2.38 | 3.43 ± 2.47 | 0.564 | 2.80 ± 1.82 | 2.87 ± 2.10 | 0.928 | 3.14 ± 1.29 | 3.50 ± 1.51 | 0.160 | 2.87 ± 1.81 | 2.87 ± 1.77 | 1.000 | 0.618 |
| MNA | 23.86 ± 3.24 | 23.64 ± 3.51 | 0.721 | 24.83 ± 3.96 | 25.67 ± 4.29 | 0.140 | 23.68 ± 4.90 | 23.29 ± 5.08 | 0.475 | 24.03 ± 3.89 | 23.90 ± 3.75 | 0.806 | 0.755 |
| **Social frailty score** | 1.50 ± 0.65 | 1.36 ± 0.63 | 0.317 | 1.60 ± 1.12 | 1.47 ± 1.46 | 0.596 | 2.29 ± 1.27 | 2.21 ± 1.25 | 0.564 | 2.53 ± 1.46 | 2.33 ± 1.35 | 0.380 | 0.142 |
| GDS15 | 3.00 ± 2.32 | 3.43 ± 3.16 | 0.472 | 3.80 ± 2.93 | 3.87 ± 4.02 | 0.543 | 4.36 ± 3.59 | 4.57 ± 2.79 | 0.478 | 5.87 ± 4.50 | 5.53 ± 4.52 | 0.561 | 0.322 |

This table presents baseline and follow-up values for frailty criteria, physical performance, sarcopenia-related measures, oral function, and nutritional status in four oral exercise frequency groups under the intention-to-treat analysis. Bold variables indicate the main outcome measures: Number of frailty criteria, SMI, Number of present teeth, Number of functional teeth, Number of oral hypofunction criteria, MNA, Social frailty score, and GDS15.

*CI* confidence interval, *MCID* minimum clinically important difference, *BMI* body mass index, *SMI* skeletal mass index, *EAT10* The 10-item Eating Assessment. Tool, *RSST* repetitive saliva swallowing test, *MNA* Mini Nutritional Assessment, *GDS* Geriatric depression scale.

[a]Wilcoxon signed-rank test at baseline and follow-up.
[b]Kruskal-Wallis test at baseline for the four groups.
[c]Average ± standard deviation.

**Table 3 | Results of two-way repeated measures ANOVA analysis in 4 groups (intention-to-treat analysis)**

| Variables | Mean difference (CI) | | | | Time | | | Group x Time | | |
|---|---|---|---|---|---|---|---|---|---|---|
| | 3 times/day & everyday | 3 times/day & 3 days/week | once/day & everyday | once/day & 3 days/week | F | p | $\eta p^2$ | F | p | $\eta p^2$ |
| Number of frailty criteria | −0.357 (−0.787, 0.073)[a] | −0.600 (−1.255, 0.055) | −0.571 (−1.379, 0.236) | −0.600 (−1.008, −0.192) | 14.803 | <0.001 | 0.215 | 0.212 | 0.888 | 0.012 |
| SMI (kg/m²) | −0.017 (−0.316, 0.282)[b] | 0.083 (−0.102, 0.268) | −0.027 (−0.134, 0.079) | 0.021 (−0.377, 0.418) | 0.121 | 0.729 | 0.002 | 0.720 | 0.545 | 0.041 |
| Number of present teeth (n) | −0.071 (−0.300, 0.157) | 0.000 (0.000, 0.000) | 0.000 (0.000, 0.000) | 0.000 (0.000, 0.000) | 1.074 | 0.305 | 0.020 | 0.043 | 0.988 | 0.002 |
| Number of functional teeth (n) | −0.071 (−0.300, 0.157) | 0.000 (0.000, 0.000) | 0.000 (0.000, 0.000) | 0.000 (0.000, 0.000) | 1.074 | 0.305 | 0.020 | 1.517 | 0.221 | 0.078 |
| Number of Oral hypofunction criteria | −0.500 (−1.304, 0.304) | −0.333 (−1.386, 0.719) | −0.643 (−1.735, 0.449) | −0.933 (−1.717, −0.150) | 16.456 | <0.001 | 0.234 | 1.769 | 0.164 | 0.090 |
| MNA | −0.214 (−2.583, 2.154) | 0.833 (−1.696, 3.363) | −0.393 (−1.949, 1.163) | −0.133 (−1.730, 1.463) | 0.005 | 0.943 | 0.000 | 0.597 | 0.620 | 0.032 |
| Social frailty score | −0.143 (−0.600, 0.314) | −0.133 (−1.194, 0.928) | −0.071 (−0.477, 0.334) | −0.200 (−0.967, 0.567) | 1.371 | 0.247 | 0.025 | 3.006 | 0.038 | 0.143 |
| GDS15 | 0.429 (−1.336, 2.193) | 0.067 (−1.869, 2.002) | 0.214 (−1.215, 1.643) | −0.333 (−2.569, 1.902) | 0.100 | 0.753 | 0.002 | 1.431 | 0.244 | 0.074 |

This table summarizes mean differences (95% confidence intervals) from baseline to follow-up for frailty criteria, sarcopenia-related measures, oral hypofunction, and other outcomes across four oral exercise frequency groups under the intention-to-treat analysis. It also presents F values, p values, and partial $\eta^2$ for the main effect of time and the interaction between group and time. hp2: partial eta squared as a measure of the effect size.

CI confidence interval, SMI skeletal mass index, MNA Mini Nutritional Assessment, GDS Geriatric depression scale.

[a]95% Confidence interval.

[b]99.3% Confidence interval.

CI, −1.379 to 0.236) in the once/day & everyday group, and −0.600 (95% CI, −1.008 to −0.192) in the once/day & 3 days/week group. A two-way repeated measures ANOVA in the ITT analysis revealed a significant main effect of time for this outcome ($F_{(1, 54)} = 14.803$, $p < 0.001$, partial $\eta^2 = 0.215$), indicating an overall reduction in frailty criteria after the intervention across all frequency groups. Bonferroni-adjusted pairwise comparisons showed no significant differences between groups (all $p > 0.007$).

### Secondary outcomes

A significant main effect of time was observed for the number of applicable items for oral hypofunction ($F(1, 54) = 16.456$, $p < 0.001$, partial $\eta^2 = 0.234$), suggesting improvement in oral function followings. The PP analysis showed similar results to the ITT analysis (see Supplementary Tables 3–5 for detailed results). An additional analysis using categorical data with a generalized linear model (logistic regression) was conducted on frailty, EAT-10, social frailty and GDS-15, and the results were consistent (see Supplementary Tables 6 and 7 for detailed ITT and PP results).

### Implementation rates

The average implementation rates and 95%CI for the groups were as follows: 3 times/day & everyday group, 66.4 ± 35.0 (%), 95%CI (42.29, 87.56); 3 times/day & 3 days/week group, 93.8 ± 15.1, 95%CI (85.07, 102.50); once/day & everyday group, 57.2 ± 38.0, 95%CI (29.99, 84.41); and once/day & 3 days/week group, 69.1 ± 41.8, 95%CI (44.93, 93.17). The results of a Kruskal-Wallis test with Bonferroni correction indicated that the 3 times/day & 3 days/week group had a significantly higher implementation rate than the 3 times/day & everyday and once/day & everyday groups.

### Possible harms

Harms were defined as any adverse events or discomfort related to the oral exercise program, such as pain, fatigue, or injury. These were assessed non-systematically: participants were instructed to report any adverse effects if they occurred during the intervention period. No adverse effects were reported throughout the conduct of the entire study.

## Discussion

In this study, the 3-month intervention of oral exercises alone resulted in a significant improvement in frailty. The frailty assessment showed similar results when analyzed using the categories of robust, pre-frailty, and frailty (Supplementary Table 4). To the best of our knowledge, such findings have not been extensively reported in previous research.

Previous studies have reported that combining oral exercises with other interventions leads to enhanced physical abilities. Watanabe et al.[36] reported improvements in physical function through a 12-week self-monitored comprehensive geriatric intervention program that consisted of four components: low-load resistance exercises using body weight and rubber bands, increased physical activity, oral function care, and nutritional guidance. Kito et al.[37] reported that a group undergoing a combined program of physical and oral exercises showed significantly greater improvements in both oral and physical functions compared with a 12-week physical exercise intervention group, but did not discuss the relationship between improvements in oral function and physical function.

In the present study, no significant temporal main effects of nutritional status or skeletal muscle mass index (SMI) were observed, which suggests that improvements in physical function were not due to increases in skeletal muscle mass through nutritional status; other factors may have been involved. As possible other factors, examination itself or self-monitoring can be involved. The previous intervention studies suggest the effects of examination on disease improvements[38] and reduction of worry[39], and the effects of self-monitoring on improvement in metabolic syndrome[40]. Thus, in this study, baseline oral examination and/or self-monitoring of oral exercise might affect improvement in frailty.

Also in this study, the implementation of oral exercises alone did not improve nutritional status. In a systematic review, Smit et al.[41] reported that

only cross-sectional studies were available regarding the relationship between masticatory function and nutritional status, and the results were not very consistent. Ozsürekci et al.[42] found no association between chewing ability and the Mini Nutritional Assessment Short Form (MNA-SF). Iwasaki et al.[43] reported that a low number of current teeth was associated with malnutrition as evaluated by the MNA-SF or serum albumin levels. In the present study, only guidance on oral exercises was provided, without any nutritional counseling. Although objective improvements in oral function were observed, the individuals did not have self-awareness of these improvements and their dietary content may not have changed. This could be one of the factors that did not lead to an improvement in nutritional status.

It has been reported that providing nutrition and oral hygiene guidance to older adults and their caregivers leads to improvements in the nutritional status of older adults[44]. Zhu et al.[45] reported that combining swallowing exercises with nutritional guidance for patients with laryngeal cancer leads to improvements in swallowing function and nutritional status. Kikutani et al.[46] found that an intervention combining high-calorie, high-protein diets with oral function training for older individuals requiring care resulted in the maintenance of oral function and improvements in nutritional status. However, to our knowledge, no studies have indicated that interventions including oral exercises do not improve nutritional status. It is possible that a multifaceted intervention that includes not only oral exercises, but also other methods related to nutritional status, could lead to improvements in nutritional status.

In this study, the implementation of oral exercises alone did not lead to improvements in sarcopenia. In a systematic review, Smit et al.[41] reported that only cross-sectional studies were available regarding the relationship between chewing function and sarcopenia, and the results were not very consistent. In a study involving community-dwelling older individuals, Yoshida et al.[47] reported a correlation between chewing ability and sarcopenia. By contrast, Aquilanti et al.[48] discussed the potential influence of a well-trained dietitian providing specialized dietary therapy to institutionalized older individuals, but reported no correlation between chewing ability and sarcopenia. Given that no improvement in nutritional status was observed in the present study, it is possible that there was also no impact on sarcopenia. Watanabe et al.[36] reported that interventions involving exercise, oral function, and nutrition improved physical function and the thickness of the anterior thigh muscles, thereby suggesting a potential prevention strategy for sarcopenia. To the best of our knowledge, no studies have indicated that interventions including oral exercises do not improve sarcopenia. It is possible that a multifaceted intervention, which includes not only oral exercises, but also other methods related to muscles, could lead to improvements in sarcopenia.

In the present study, the frequency of three times a day & 3 days a week had the highest implementation rate. Shu et al.[49] reported that the frequency and duration of exercises varied among the reviewed studies in their systematic review on oral exercises and the improvement of oral function. In the subgroup meta-analysis,[49] they showed that both high- and low-intensity daily exercises were effective for the improvement of oral function; however, they did not discuss adherence rates. They[49] also noted that the improved occlusal force after ceasing exercise is expected to be maintained for about 2–3 weeks, gradually decreasing to pre-intervention levels. Therefore, similar to other forms of physical exercise, regular oral exercises are necessary to sustain such effects[50,51]. Accordingly, the frequency of three times a day & 3 days a week may be effective for continuing long-term exercise. Adherence rates may be also influenced by factors other than the ease of frequency such as participants' experiences, difficulties, and preferences. However, this study did not conduct qualitative assessments of these factors, leaving the impact on adherence rates unclear. To more accurately examine adherence rates, it is necessary to consider the characteristics of the participants. Additionally, during the 3-month study period, we examined adherence at a uniform frequency from start to finish, and found that a frequency of three times a day, three times a week, had the highest implementation rate. For example, a more flexible approach that starts with fewer sessions in the initial weeks and then transitions to daily implementation might lead to higher adherence rates in the long term.

In intervention research, it is essential to recognize that identifying effective intervention methods based on evidence is just the beginning; the real significance lies in their actual social implementation. The findings of the present study suggest that oral exercises not only contribute to frailty improvement, but also propose a feasible implementation frequency of three times a day & 3 days a week. This aspect aligns with implementation research[52], bridging the gap between evidence-based interventions and their practical application in society.

The strength of this study lies in its examination of the effects of oral exercises solely on the improvement of frailty. Additionally, this study investigated feasible implementation frequencies and included aspects of implementation research, which is the next phase following intervention studies.

There are several limitations in this study. The first limitation is the lack of a control group in this study. In a previous study that evaluated the improvement of frailty using a mobile/web-based application and sensor kits, only 9.9% of the control group (pre-frailty, $n = 44$, average age 82.6 years) became robust after 3 months[53]. In another study that assessed the impact of an educational program on the frailty status, physical function, physical activity, sleep patterns, and nutritional status of older adults living in the community, none of the control group (pre-frailty and frailty, 83 participants with an average age of 74.9 years) became healthy after 6 months[54]. Additionally, a study focusing on older adults who were frailty or pre-frailty in the community evaluated the effects of an educational program on depressive symptoms, cognitive function, social support, quality of life, and physical frailty. The median number of frailty criteria for the control group (pre-frailty, n = 90, average age 75.7 years) in the intervention study remained unchanged at after 6 months[55]. Although simple comparisons cannot be made due to differences in community and background, it can be inferred that the frailty status is unlikely to improve significantly without any intervention. Second, the participants were restricted to patients regularly visiting a university hospital's preventive dentistry department. It is possible that the participants had a higher health consciousness than the general population, which may have resulted in an adherence rate that was higher than what would typically be expected. They could have been proactive and cooperative regarding treatment. Therefore, a broader range of subjects needs to be investigated. Third, the reevaluation period of 3 months was relatively short. Kito et al.[37] reported that a 12-week physical and oral exercise program effected oral and physical function in older individuals. Watanabe et al.[36] reported that a month oral exercise effected 12 week physical and oral exercise increased in daily steps. Koponen et al.[44] reported that the intervention group receiving individual nutritional counseling and oral health guidance for 6 months showed an improvement in nutritional status. To investigate whether oral exercises alone influence the improvement of frailty, it may have been necessary to establish an intervention period that accounts for the duration required for changes in nutritional status, sarcopenia status, and psychosocial conditions, which are intervening factors. A more comprehensive study would benefit from a longer research duration, such as 6 months. Fourth, the implementation rates for the two groups that performed oral exercises daily were low. This may have been due to participants feeling burdened by the daily requirement, leading to a decrease in implementation rates. Fifth the intervention methods do not include exercise or nutrition. While it has been reported that the combination of exercise and nutrition is beneficial for frailty prevention[56], this study aimed to investigate the impact of oral exercises on frailty, and therefore, interventions related to exercise and nutrition could not be examined. Further investigation is needed.

## Conclusion

After conducting oral exercises for 3 months on older adults with pre-frailty or frailty, improvements in frailty were observed within the aforementioned limitations. The implementation frequency that yielded the best results was

three times/day, 3 days/week for older people without disabilities or cognitive impairments.

## Data availability

Due to the study protocol approved by the ethics committee, which specifies that participant data will not be reused for secondary purposes, anonymized individual-level data cannot be shared beyond what is presented in this article. All aggregated results are included in the figures and tables within the manuscript, and the numerical source data underlying the tables are provided in Supplementary Data 1. The study protocol is publicly available at the Japan Registry of Clinical Trials (jRCT1062210063).

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

## Acknowledgements
This work was supported by a KAKEN Grant-in-Aid from the Japan Society for the Promotion of Science (21K10208). This research did not receive any other specific grants from funding agencies in commercial or not-for-profit sectors. The authors thank FORTE Science Communications (https://www.forte-science.co.jp/) for English language editing.

## Author contributions
N.T.: Conceptualization, Methodology, Data curation, Formal analysis, Writing—original draft, Visualization, Project administration, Funding acquisition. N.S.: Data curation, Investigation, Writing—review and editing, Visualization. S.I.: Data curation, Investigation, Writing—review and editing. M.M.: Conceptualization, Methodology, Formal analysis, Writing—review and editing. D.E.: Conceptualization, Methodology, Formal analysis, Writing—review and editing. All authors read and approved the final manuscript.

## Competing interests
The authors declare no competing interests.

## Additional information

**Noriko Takeuchi** [ID][1] ✉**, Nanami Sawada[2], Sakura Inada[3], Manabu Morita[4] & Daisuke Ekuni[5]**

---

[1]Department of Preventive Dentistry, Division of Dentistry, Medical Development Field, Okayama University, Okayama, Japan. [2]Section of Preventive and Public Health Dentistry, Division of Oral Health, Growth and Development, Faculty of Dental Science, Kyushu University, Fukuoka, Japan. [3]Division of Health Promotion, Okayama-City Health Center, Okayama, Japan. [4]Department of Oral Health Sciences, Faculty of Health Care Sciences, Takarazuka University of Medical and Health Care, Hyogo, Japan. [5]Department of Preventive Dentistry, Faculty of Medicine, Dentistry and Pharmaceutical Sciences, Okayama University, Okayama, Japan.
✉e-mail: takeuti@md.okayama-u.ac.jp

