## [Transparent Peer Review file · Communications Medicine]

Effects of an oral exercise intervention on pre-frailty or frailty in older people: a randomized clinical trial

Corresponding Author: Dr Noriko Takeuchi

Version 0:

Reviewer comments:

Reviewer #1

(Remarks to the Author)

The study addresses an important and underexplored area in geriatric oral health, particularly the role of oral exercise in mitigating frailty. The randomized design strengthens the validity of the findings. However, there are certain methodological concerns and areas requiring clarification to enhance the robustness of the conclusions. The study's implications for clinical practice and public health are noteworthy, but further refinements in data analysis and discussion are needed.

Study Design & Methodology:

Blinding is mentioned but not explicitly detailed regarding how assessors were blinded and whether participants were aware of their group assignment, potentially introducing bias. Please add that although this study states that the evaluators were not informed of which group the subjects were assigned to, only one research team member maintained information about study group assignment and communication with subjects occurred only through that one person.

Intervention Feasibility and Adherence:

The study highlights that the three times/day, three days/week group had the highest adherence rate (93.8%). However, reasons for higher compliance should be explored further.

Were there any qualitative assessments regarding participant experiences, difficulties, or preferences? These insights could improve real-world implementation.

In the present study, the frequency of three times a day, three days a week had the highest implementation rate. The researchers said that a frequency of three times a day, three days a week could be effective when operating in the long term. However, in reality, the subjects may have felt burdened by exercising every day while participating in the study, and it may be a better strategy to have an adaptation period with a small number of times in the beginning and then perform it every day in the long term. Therefore, I think that additional consideration is needed in this area.

Limitations:

The study appropriately acknowledges its limitations, such as the single-center design and relatively short intervention period. However, additional discussion on potential selection bias (e.g., participants recruited from a university hospital) would be valuable.

The authors suggest that future studies should extend the duration of interventions. Providing a more specific recommendation on the optimal timeframe would strengthen this point.

Reviewer #2

(Remarks to the Author)

Thank you for the opportunity to review this article for publication in Communications medicine.

This randomized controlled trial is an important study for improving oral functions in older adults with frailty and determining

the appropriate frequency of oral exercises.

Setting a feasible and evidence-based number of training periods is highly useful for implementation in clinical settings. As mentioned in the limitations of this study, the subjects were older adults who regularly visited the preventive dentistry department of a university hospital and were highly interested in oral care and health maintenance. In order to approach health maintenance for a broader population in the community, methods that take into account social background and other factors are required.

Comment for over all :

There does not seem to be a specified format for manuscripts, but the overall structure of the paper is difficult to understand. The "Participant Flow" and "Recruitment" sections are recommended to be moved from "Results" to "Materials and Methods."

Intervention studies are being conducted in accordance with the consort checklist. Statistical methods also appear to be appropriate.

Reviewer #3

(Remarks to the Author)

This paper reports results from a randomized clinical trial comparing the effects oral exercise regimens on frailty and other outcomes.

The introduction sets the stage well, but cutting right to the chase, the lack of a control group is a major problem with this trial. Without it, you're able to compare the effect of different oral exercise regimens on frailty and other outcomes, but you're not able to compare any of those effects to what would happen without a regimen of this sort in place. Without that comparison, it's hard for a clinician to know whether prescribing the exercise at all is worthwhile.

There's also a general problem with how the results are presented. None of the actual differences, or even the directions of the differences, are reported in the manuscript, and no minimal clinically important differences (MCIDs) are provided in the descriptions of the outcomes. This information is necessary for a clinical audience to assess whether the differences are clinically meaningful rather than simply statistically significant.

More specific questions and comments follow.

METHODS

At line 221, "Based on..." makes it sound like a stratified randomization. I think you want something like, "Participants were randomly assigned to one of four exercise frequencies:"

Was there a statistical analysis plan with pre-specified analyses written before the data were unblinded? If so, it should be included in the supplementary materials.

Were baseline data collected before or after randomization?

For each outcome, the MCID should be given (with reference); if one hasn't been established for an outcome, that should be noted.

Did each assessor only assess outcomes from one treatment group (lines 400-401 imply the assessor is assigned to one group but doesn't know which group that is)? If so, that introduces a confound between treatment and assessors.

Was study leadership blinded to the outcome data until the primary and secondary outcomes were analyzed?

Several outcomes -- frailty, mental health status, social frailty, and swallowing function -- have cutpoints defining clinical categories. ANOVA (and the general linear model that underlies it) assumes that a 1-point change in outcome is equally meaningful anywhere along the outcome scale, but that doesn't hold if a change that crosses a threshold is more clinically relevant than an equal-sized change that sits within a clinical category. In that situation, analyzing the categorized (binary) data with a generalized linear model (logistic regression is the classic choice and generates odds ratios, though risk ratios, which are more intuitive to interpret, can be obtained by robust Poisson regression with a sandwich variance estimator) is often a more appropriate choice. For these outcomes, are the continuous scores or the categories more clinically relevant? If the ANOVA was pre-specified, the binary outcomes should be done as a sensitivity analysis to see if the conclusions would have been the same.

Did correction for multiple comparisons account for the number of secondary outcomes as well as the number of comparisons within each outcome? (The primary outcome doesn't need to be corrected for the presence of the secondary outcomes, but the full set of secondary analyses should be held to an overall .05 type I error rate [FWER].)

RESULTS

Table 2 is about baseline and follow-up outcomes, not participant characteristics.

Since the comparisons in Table 3 are between two means, the difference in means with a confidence interval (95% CI for the primary outcome; the 22 secondaries should have a CI of $1 - .05/22 = 99.8\%$) should be presented along with the partial eta-squared. To clinicians, the differences are more relevant than the proportions of variance explained.

The text and abstract of the paper should provide the means that are being compared and their difference, with confidence intervals, for each 1-df comparison, while Fs can be relegated to the tables. Again, it is the differences that are of clinical interest, and not all statistically significant differences are clinically meaningful.

For implementation, we need to see rates and rate ratios with confidence intervals. A Poisson regression, with the number of times exercises were performed as the outcome and the number of times exercises should have been performed as a log offset, will provide rate ratios and probably be more powerful than the U test.

DISCUSSION

While a lot of the material in the discussion is good, the fundamental problem is that while there is improvement over the 3 months of exercise, and the amount of improvement varies with exercise regimen, there are no data to show that these outcomes are better than what would be observed with no exercise regimen in place.

Reviewer #4

(Remarks to the Author)

This RCT is well designed to investigate if oral exercises can improve pre-frailty and frailty. Given that frailty is multi-dimensional and guidelines have repeatedly suggested the use of combined exercise and nutrition, expecting that oral exercises alone can improve pre-frailty and frailty seems like a tall order.

Nonetheless, much work has gone into the study and the results should be shared with the scientific community. Though, this study needs a major+ revision, which is better as a resubmission rather than just a revision.

I have a few suggestions:

major:

1. invite another author with significant trial experience, especially in effectiveness:implementation design which is what this trial was about, to help restructure the presentation of this study. For example, present the primary finding which was insignificant, then pool the secondary outcomes together as a section. Afterwhich, share about implementation findings e.g., adherence.

minor

2. please rename the groups - the use of numbers/alphabets makes it really hard to read.

3. The implementation rates

4. do not overstate your findings. the study participants were able to walk independently, and respond to the questionnaires. Please ensure that the results and conclusions highlight the fact that these are older adults that are ?without disabilities or cognitive impairments? The results was only significant for social frailty. This needs to be highlighted through the results where applicable.

Version 1:

Reviewer comments:

Reviewer #1

(Remarks to the Author)

I appreciate the authors' thoughtful and comprehensive responses to the initial review. The revised manuscript addresses the key concerns I raised in a clear and satisfactory manner. Overall, I find that the revised manuscript is significantly improved. The authors have engaged constructively with the feedback, and I believe the paper now offers a more complete and clinically relevant contribution to the field of geriatric oral health.

I support the publication of this article following final editorial checks.

Reviewer #5

(Remarks to the Author)

Noriko et al performed an RCT to assess the effect of an oral exercise intervention on frailty for older people. I have the following concerns;

- 1, The current title is ambiguous. Do you mean (a) two separate outcomes ("pre-frailty" and "frailty"), or (b) a study population

that includes older people with either pre-frailty or frailty? Please clarify the intended meaning and revise the title to remove ambiguity (for example: "...in older adults with pre-frailty or frailty" or "...on progression from pre-frailty to frailty").

2, It is unclear whether this trial is factorial (e.g., two factors: everyday and once/day) or a parallel multi-arm trial. If the intervention has two orthogonal factors, a factorial design should be considered and explicitly justified. If not factorial, please justify the chosen design and explain why interactions between dosing/frequency factors are not being modelled.

3, Please state the primary outcome clearly and early (Methods and Abstract). Results for the primary outcome should be presented with effect estimates and 95% confidence intervals comparing randomised groups (not only p-values). Include the exact estimand (e.g., difference in mean score, odds ratio for progression) and the analysis population used.

4, The sample size calculation is said to target an effect size of 0.70 — please specify which outcome this refers to (primary outcome) and exactly how the effect size was defined (Cohen's d, ratio, absolute difference). Clarify whether the same sample size calculation applies to key secondary outcomes (usually it does not) and, if any secondary outcome was used to power the study, make that explicit.

5, The manuscript states results were presented after considering multiplicity — please clarify: (a) which comparisons/outcomes had multiplicity adjustments, (b) what adjustment method was prespecified (e.g., Bonferroni, Holm, hierarchical testing), and (c) whether multiplicity handling was prespecified in the protocol/SAP. Given four groups, adjustments limited to a subset of outcomes may be insufficient; please provide the prespecified strategy for handling multiplicity across multiple groups and multiple outcomes.

6. In accordance with CONSORT, present baseline characteristics by randomised group in Table 1 (not pooled). This allows readers to evaluate the success of randomisation and baseline balance.

Version 2:

Reviewer comments:

Reviewer #5

(Remarks to the Author)

Thanks for providing the responses. This version has been improved a lot. However, I still have the following concerns.

1, the revised title does not match with the responses. Please ensure this is correctly changed

2, the author did not clarify the reasons for not including the interaction.

Version 3:

Reviewer comments:

Reviewer #5

(Remarks to the Author)

Thanks for providing the response. I have no further comments

Reviewers' comments:

Reviewer #1 (Remarks to the Author):

The study addresses an important and underexplored area in geriatric oral health, particularly the role of oral exercise in mitigating frailty. The randomized design strengthens the validity of the findings. However, there are certain methodological concerns and areas requiring clarification to enhance the robustness of the conclusions. The study's implications for clinical practice and public health are noteworthy, but further refinements in data analysis and discussion are needed.

Study Design & Methodology:

Blinding is mentioned but not explicitly detailed regarding how assessors were blinded and whether participants were aware of their group assignment, potentially introducing bias. Please add that although this study states that the evaluators were not informed of which group the subjects were assigned to, only one research team member maintained information about study group assignment and communication with subjects occurred only through that one person.

Our comments: Thank you for your suggestions. ~~We have added to the manuscript.~~ We have added the following to the manuscript: “The evaluators were not informed of which group the subjects were assigned to, only one research team member maintained information about study group assignment and communication with subjects occurred only through that one person.” (L441-444)

Intervention Feasibility and Adherence:

The study highlights that the three times/day, three days/week group had the highest adherence rate (93.8%). However, reasons for higher compliance should be explored further. Were there any qualitative assessments regarding participant experiences, difficulties, or preferences? These insights could improve real-world implementation.

Our comments: Thank you for your suggestion. There was no qualitative assessment of participants' experiences, difficulties, or preferences that might affect compliance rates

in this study. We have added to the discussion as follows ; “Adherence rates may be also influenced by factors other than the ease of frequency such as participants' experiences, difficulties, and preferences. However, this study did not conduct qualitative assessments of these factors, leaving the impact on adherence rates unclear. To more accurately examine adherence rates, it is necessary to consider the characteristics of the participants.” (L164-167) .

In the present study, the frequency of three times a day, three days a week had the highest implementation rate. The researchers said that a frequency of three times a day, three days a week could be effective when operating in the long term. However, in reality, the subjects may have felt burdened by exercising every day while participating in the study, and it may be a better strategy to have an adaptation period with a small number of times in the beginning and then perform it every day in the long term. Therefore, I think that additional consideration is needed in this area.

Our comments: Thank you for your suggestion. During the three-month study period, we examined adherence at a uniform frequency from start to finish, and found that a frequency of three times a day, three times a week, had the highest implementation rate. As you pointed out, a more flexible approach, such as starting with fewer sessions in the initial weeks and then moving to daily implementation, may lead to higher adherence rates. These considerations have been added to the discussion. (L167-171)

Limitations:

The study appropriately acknowledges its limitations, such as the single-center design and relatively short intervention period. However, additional discussion on potential selection bias (e.g., participants recruited from a university hospital) would be valuable. The authors suggest that future studies should extend the duration of interventions. Providing a more specific recommendation on the optimal timeframe would strengthen this point.

Our comments: Thank you for your suggestion.

Regarding participants who regularly visit a university hospital, it is possible that they have a higher health consciousness compared to the general local population. This may

have contributed to an adherence rate that was higher than what would typically be expected. This has been added to the limitations section. (L196-198)
Additionally, we have included specific durations regarding the optimal extension of the intervention period. (L200-209)

Reviewer #2 (Remarks to the Author):

Thank you for the opportunity to review this article for publication in Communications medicine.

This randomized controlled trial is an important study for improving oral functions in older adults with frailty and determining the appropriate frequency of oral exercises. Setting a feasible and evidence-based number of training periods is highly useful for implementation in clinical settings.

As mentioned in the limitations of this study, the subjects were older adults who regularly visited the preventive dentistry department of a university hospital and were highly interested in oral care and health maintenance. In order to approach health maintenance for a broader population in the community, methods that take into account social background and other factors are required.

Our comments: Thank you for your suggestions. We received feedback from other reviewers suggesting that a qualitative assessment of participants' experiences, challenges, preferences, and other background factors should be considered. Since this study did not evaluate the social backgrounds related to experiences, challenges, and preferences, it remains unclear whether they had any impact. We have added these points to the discussion. (L164-167)

Comment for over all:

There does not seem to be a specified format for manuscripts, but the overall structure of the paper is difficult to understand. The “Participant Flow” and “Recruitment” sections are recommended to be moved from “Results” to “Materials and Methods.”

Our comments: Thank you for your suggestions. The “Participant Flow” and “Recruitment” has been moved from “Results” to “Materials and Methods.” (L221-227, L229-231)

Intervention studies are being conducted in accordance with the consort checklist. Statistical methods also appear to be appropriate.

Our comments: Thank you for your comments.

Reviewer #3 (Remarks to the Author):

This paper reports results from a randomized clinical trial comparing the effects oral exercise regiments on frailty and other outcomes.

The introduction sets the stage well, but cutting right to the chase, the lack of a control group is a major problem with this trial. Without it, you're able to compare the effect of different oral exercise regimens on frailty and other outcomes, but you're not able to compare any of those effects to what would happen without a regimen of this sort in place. Without that comparison, it's hard for a clinician to know whether prescribing the exercise at all is worthwhile.

Our comments: Thank you for your suggestions.

The lack of a control group, as you point out, is a major problem with this study. We have added information regarding participants who were not subjected to intervention from previous studies to the discussion. (L183-195)

There's also a general problem with how the results are presented. None of the actual differences, or even the directions of the differences, are reported in the manuscript, and no minimal clinically important differences (MCIDs) are provided in the descriptions of the outcomes. This information is necessary for a clinical audience to assess whether the differences are clinically meaningful rather than simply statistically significant.

Our comments: Thank you for your suggestions. Clinically significant minimal differences and direction of differences have been added to the results. (L81-83, L88-89, L456-458, Table 2, Supplemental Table 2)

More specific questions and comments follow.

METHODS

At line 221, “Based on...” makes it sound like a stratified randomization. I think you want something like, “Participants were randomly assigned to one of four exercise

frequencies:"

Our comments: Thank you for your suggestions. We have revised it to, "Participants were randomly assigned to one of four exercise frequencies." (L262).

Was there a statistical analysis plan with pre-specified analyses written before the data were unblinded? If so, it should be included in the supplementary materials.

Our comments: Thank you for your suggestions. A statistical analysis plan with pre-specified analyses written before the data were unblinded is included in the clinical trial study plan (supplemental material).

Were baseline data collected before or after randomization?

Our comments: Thank you for your suggestions. Baseline data were collected before randomization. Figure 1(flow chart) were revised.

For each outcome, the MCID should be given (with reference); if one hasn't been established for an outcome, that should be noted.

Our comments: Thank you for your suggestions. Minimal clinically important differences were added to the results. (L82-83, L456-458, Table 2, Supplemental Table 2)

Did each assessor only assess outcomes from one treatment group (lines 400-401 imply the assessor is assigned to one group but doesn't know which group that is)? If so, that introduces a confound between treatment and assessors.

Our comments: Thank you for your suggestions. Each assessor evaluated regardless of group assignment. We have revised the manuscript. (L441-444)

Was study leadership blinded to the outcome data until the primary and secondary outcomes were analyzed?

Our comments: Thank you for your suggestions. Study leaders were blinded to outcome data until the primary and secondary outcomes were analyzed.

Several outcomes — frailty, mental health status, social frailty, and swallowing function — have cutpoints defining clinical categories. ANOVA (and the general linear model that underlies it) assumes that a 1-point change in outcome is equally meaningful anywhere along the outcome scale, but that doesn't hold if a change that crosses a threshold is more clinically relevant than an equal-sized change that sits within a clinical category. In that situation, analyzing the categorized (binary) data with a generalized linear model (logistic regression is the classic choice and generates odds ratios, though risk ratios, which are more intuitive to interpret, can be obtained by robust Poisson regression with a sandwich variance estimator) is often a more appropriate choice. For these outcomes, are the continuous scores or the categories more clinically relevant? If the ANOVA was pre-specified, the binary outcomes should be done as a sensitivity analysis to see if the conclusions would have been the same.

Our comments: Thank you for your suggestions. An additional analysis using binary data with a generalized linear model (logistic regression) was conducted on frailty, GDS-15, social frailty, and EAT-10. The results were consistent with those obtained from continuous variables. The table has been added to the supplemental material. (L91-93, L106-107, Supplemental Table 4, Supplemental Table 5)

Did correction for multiple comparisons account for the number of secondary outcomes as well as the number of comparisons within each outcome? (The primary outcome doesn't need to be corrected for the presence of the secondary outcomes, but the full set of secondary analyses should be held to an overall .05 type I error rate [FWER].)

Our comments: Thank you for your suggestions. In Table 2, we have corrected for comparisons of 22 items ($0.05/22=0.002$). In Table 3, we have corrected for multiple

comparisons of 7 items ($0.05/7=0.007$). (L23, L25, L85-87, L88, L454-456)

RESULTS

Table 2 is about baseline and follow-up outcomes, not participant characteristics.

Our comments: Thank you for your suggestions. We have revised the title. (Table 2).

Since the comparisons in Table 3 are between two means, the difference in means with a confidence interval (95% CI for the primary outcome; the 22 secondaries should have a CI of $1-.05/22 = 99.8\%$) should be presented along with the partial eta-squared. To clinicians, the differences are more relevant than the proportions of variance explained.

Our comments: Thank you for your suggestions. Confidence intervals (95% CI for primary outcomes; $1-.05/22 = 99.8\%$ CI for 22 secondary outcomes) were added for differences in means. (L85-87, L454-456, Table 3)

The text and abstract of the paper should provide the means that are being compared and their difference, with confidence intervals, for each 1-df comparison, while Fs can be relegated to the tables. Again, it is the differences that are of clinical interest, and not all statistically significant differences are clinically meaningful.

Our comments: Thank you for your suggestions. We have added the mean differences and confidence intervals. (L25, L85-87, L94, L95-96, L97, Table 1, Table 2, Table 3)

For implementation, we need to see rates and rate ratios with confidence intervals. A Poisson regression, with the number of times exercises were performed as the outcome and the number of times exercises should have been performed as a log offset, will provide rate ratios and probably be more powerful than the U test.

Our comments: Thank you for your suggestion. In this study, we conducted experiments at various frequencies to evaluate which frequency can be sustained for a

longer period. Therefore, we have chosen the current testing method.

DISCUSSION

While a lot of the material in the discussion is good, the fundamental problem is that while there is improvement over the 3 months of exercise, and the amount of improvement varies with exercise regimen, there are no data to show that these outcomes are better than what would be observed with no exercise regimen in place.

Our comments: Thank you for your suggestions. The lack of a control group, as you point out, is a major problem with this study. We have added information regarding participants who were not subjected to intervention from previous studies to the discussion. (L183-195)

Reviewer #4 (Remarks to the Author):

This RCT is well designed to investigate if oral exercises can improve pre-frailty and frailty. Given that frailty is multi-dimensional and guidelines have repeatedly suggested the use of combined exercise and nutrition, expecting that oral exercises alone can improve pre-frailty and frailty seems like a tall order.

Nonetheless, much work has gone into the study and the results should be shared with the scientific community. Though, this study needs a major revision, which is better as a resubmission rather than just a revision.

Our comments: Thank you for your suggestions. As mentioned, it has been reported that the combination of exercise and nutrition is beneficial for frailty prevention; however, since this study aimed to investigate the impact of oral exercises on frailty, interventions related to exercise and nutrition could not be examined. This has been added to the limitations and conclusion section. (L210-213, L217)

I have a few suggestions:

major:

1. invite another author with significant trial experience, especially in effectiveness:implementation design which is what this trial was about, to help restructure the presentation of this study. For example, present the primary finding which was insignificant, then pool the secondary outcomes together as a section. After which, share about implementation findings e.g., adherence.

Our comments: Thank you for your suggestions. The manuscript and tables were revised based on various suggestions from reviewer 3.

We have added a discussion on the absence of a non-intervention control group (L183-195), the direction of the differences (L88-89), minimal clinically important differences (L82-83, L456-458), and the confidence intervals for the mean differences (L25, L85-87, L94, L95-96, L97, Table 1, Table 2, Table 3).

For the categorized indicators (frailty, GDS-15, social frailty, and EAT-10), We have included an analysis using binary data with a generalized linear model (logistic regression). (L91-93, L106-107, Supplemental Table 4, Supplemental Table 5)

Regarding the secondary outcomes, We considered the number of comparisons within each outcome, ensuring that the overall secondary analyses maintain a type I error rate

(FWER) of 0.05. (L23, L25, L85-87, L88, L454-456)

minor

2. please rename the groups - the use of numbers/alphabets makes it really hard to read.

Our comments: Thank you for your suggestions. We have revised them to “3 times/day & everyday” group, “3 times/day & 3 days/week” group, “once/day & everyday” group and “once/day & 3 days/week” group. (L19-20, L24-25, L26-27, L94-95, L96, L98-99, L155, L162, L175, L225-227, L262-263, L436-438, L459-460)

3. The implementation rates

Our comments: We have not been able to respond because we do not understand the content of your comments.

4. do not overstate your findings. the study participants were able to walk independently, and respond to the questionnaires. Please ensure that the results and conclusions highlight the fact that these are older adults that are ?without disabilities or cognitive impairments? The results was only significant for social frailty. This needs to be highlighted through the results where applicable.

Our comments: Thank you for your suggestions. We have added that the results of this study pertain to older adults that are without disabilities or cognitive impairments. (L27-28, L90-91, L218)

Thank you to the reviewers for their peer review.

We have revised the manuscript and the tables regarding your suggestions as follows:

Reviewers' comments:

Reviewer #1 (Remarks to the Author):

I appreciate the authors' thoughtful and comprehensive responses to the initial review. The revised manuscript addresses the key concerns I raised in a clear and satisfactory manner. Overall, I find that the revised manuscript is significantly improved. The authors have engaged constructively with the feedback, and I believe the paper now offers a more complete and clinically relevant contribution to the field of geriatric oral health.

I support the publication of this article following final editorial checks.

Our comments: Thank you for your comments. Your valuable insights and constructive feedback have helped improve the quality and clarity of the paper.

Reviewer #5 replacement referee (Remarks to the Author):

Noriko et al performed an RCT to assess the effect of an oral exercise intervention on frailty for older people. I have the following concerns;

1, The current title is ambiguous. Do you mean (a) two separate outcomes (“pre-frailty” and “frailty”), or (b) a study population that includes older people with either pre-frailty or frailty? Please clarify the intended meaning and revise the title to remove ambiguity (for example: “...in older adults with pre-frailty or frailty” or “...on progression from pre-frailty to frailty”).

Our comments: Thank you for your suggestions. The intended meaning is (b). The study population includes older adults with either pre-frailty or frailty. To clarify this, the title has been revised. (L2)

2, It is unclear whether this trial is factorial (e.g., two factors: everyday and once/day) or a parallel multi-arm trial. If the intervention has two orthogonal factors, a factorial design should be considered and explicitly justified. If not factorial, please justify the chosen design and explain why interactions between dosing/frequency factors are not being modelled.

Our comments: Thank you for your suggestions. This trial was designed as a prospective, parallel multi-arm randomized controlled trial rather than a factorial design. All intervention groups received the same oral exercise program, and the only difference among the four groups was the frequency of implementation. Because the intervention did not involve two independent orthogonal factors (e.g., dosing and frequency), a factorial design was not applicable. (L84, L85-86)

3, Please state the primary outcome clearly and early (Methods and Abstract). Results for the primary outcome should be presented with effect estimates and 95% confidence intervals comparing randomised groups (not only p-values). Include the exact estimand (e.g., difference in mean score, odds ratio for progression) and the analysis population used.

Our comments: Thank you for your suggestions. We have revised the Abstract, Methods, and Results sections to improve clarity regarding the primary outcome and the presentation of effect estimates. In the Abstract, we explicitly stated the primary outcome as the change in the number of frailty criteria from baseline to follow-up and provided the effect estimates with 95% confidence intervals. In the Methods section, we clarified that the primary analysis was conducted on the intention-to-treat (ITT) population, with per-protocol (PP) analysis as a sensitivity analysis. We also specified the statistical methods used, including Bonferroni correction for multiple comparisons and the confidence interval approach (95% CI for the primary outcome and Bonferroni-adjusted CIs for secondary outcomes). (L19-20, L24-31, L331-332, L333-335, L342-344, L355-357, L385, L392-400, L403-404)

4, The sample size calculation is said to target an effect size of 0.70 — please specify which outcome this refers to (primary outcome) and exactly how the effect size was defined (Cohen's d, ratio, absolute difference). Clarify whether the same sample size calculation applies to key secondary outcomes (usually it does not) and, if any secondary outcome was used to power the study, make that explicit.

Our comments: Thank you for your suggestions.

The sample size calculation was based on the primary outcome, namely improvement in frailty after oral exercise intervention. As no previous studies on oral exercise alone for frailty were available, reference values were obtained from studies reporting changes in tongue pressure. Effect sizes were defined as Cohen's d for change scores and converted to Cohen's f for repeated-measures ANOVA. Using $\alpha = 0.01$, power = 0.95, within-subject correlation $r = 0.5$, and the error variance ($\text{var}_{\text{error}}$) derived from pooled SD, 12 participants per group were required. Secondary outcomes were not used for power calculations. (L112-122, L153-154)

5, The manuscript states results were presented after considering multiplicity — please clarify: (a) which comparisons/outcomes had multiplicity adjustments, (b) what adjustment method was prespecified (e.g., Bonferroni, Holm, hierarchical testing), and (c) whether multiplicity handling was prespecified in the protocol/SAP. Given four groups, adjustments limited to a subset of outcomes may be insufficient; please provide the prespecified strategy for handling multiplicity across multiple groups and multiple outcomes.

Our comments: Thank you for your suggestions. We have revised the Statistical Analysis section to clarify the strategy for handling multiplicity. Specifically, Bonferroni correction was applied to all planned pairwise comparisons among the four groups for both primary and secondary outcomes to control the family-wise error rate. For the primary outcome, 95% confidence intervals were used, whereas Bonferroni-adjusted confidence intervals (e.g., 99.3% or 99.8%) were applied for secondary outcomes to account for multiplicity. This adjustment was not prespecified in the protocol or SAP; however, it was implemented post hoc to ensure rigorous control of type I error across multiple comparisons. (L355-359)

6. In accordance with CONSORT, present baseline characteristics by randomised group in Table 1 (not pooled). This allows readers to evaluate the success of randomisation and baseline balance.

Our comments: Thank you for your suggestions. We have revised Table 1 to present baseline characteristics by randomized group, in accordance with CONSORT guidelines. (Table 1, L779-780, Supplementary table 1)

Revisions to sections pointed out by reviewers are marked with yellow highlighters, while revisions to sections pointed out by the journal are marked with light blue highlighters. An error was identified in Supplementary Table 2, so it has been corrected in the revised version (highlighted in red).

Thank you to the reviewers for their peer review.

We have revised the manuscript and the tables regarding your suggestions as follows:

Reviewers' comments:

Reviewer #5 (Remarks to the Author):

Thanks for providing the responses. This version has been improved a lot. However, I still have the following concerns.

1, the revised title does not match with the responses. Please ensure this is correctly changed

Our comments: Thank you for your comments. We have revised the title to "...pre-frailty or frailty..." and changed it from "...pre-frailty and frailty...". In addition, we have revised all occurrences in the main text to reflect this correction. (L2, L44-45, L123)

2, the author did not clarify the reasons for not including the interaction.

Our comments: Thank you for your comment. In this study, the terms, "dose" and "frequency" in your comments refer to the number of oral motor sessions, "per day" and "per week", respectively. The four intervention groups were designed not only to examine the effect of oral exercises on pre-frailty or frailty improvement, but also to explore feasible implementation strategies for social application. Therefore, the interaction effects (the number of oral motor sessions, per day and per week) were less important than our purpose. We have revised Study design section. (L100-103)